# Monitoring Large-Scale Restoration Interventions from Land Preparation to Biomass Growth in the Sahel

**Moctar Sacande** \*, **Antonio Martucci and Andreas Vollrath**

Food and Agriculture Organization of the United Nations, Forestry Division, 00153 Rome, Italy;
Antonio.Martucci@fao.org (A.M.); Andreas.Vollrath@fao.org (A.V.)
\* Correspondence: Moctar.Sacande@fao.org

**Abstract:** In this work we demonstrate that restoration interventions in arid to semi-arid landscapes can be independently assessed by remote sensing methods throughout all phases. For early verification, we use Sentinel-1 radar imagery that is sensitive to changes in soil roughness and thus able to rapidly detect disturbances due to mechanised ploughing, including identification of the time of occurrence and the surface area prepared for planting. Subsequently, time series of the normalized difference vegetation index (NDVI) derived from high-resolution imagery enabled tracking and verifying of the increase in biomass and the long-term impact of restoration interventions. We assessed 111 plots within the Great Green Wall area in Burkina Faso, Niger, Nigeria and Senegal. For 58 plots, the interventions were successfully verified, corresponding to an area of more than 7000 ha of degraded land. Comparatively, these computerised data were matched with field data and high-resolution imagery, for which the NDVI was used as an indicator of subsequent biomass growth in the plots. The trends were polynomial and presented clear vegetation gains for the monthly aggregates over the last 2 years (2018–2020). The qualitative data on planted species also showed an increase in biodiversity as direct sown seeds of a minimum of 10 native Sahel species (six woody mixed with four fodder herbaceous species) were planted per hectare. This innovative and standardised monitoring method provides an objective and timely assessment of restoration interventions and will likely appeal more actors to confidently invest in restoration as a part of zero-net climate mitigation.

**Keywords:** restoration interventions; independent assessment; Great Green Wall; Sahel; remote sensing



## 1. Introduction

Large-scale restoration interventions are the priority actions set by most countries in the Sahel in their commitments for adaptation and mitigation to climate change. Restoring ecosystems is a complex undertaking. The scientific community, although widely in agreement that restoring land and ecosystems is a key strategy to avert climate change and biodiversity loss, is struggling to reach a consensus on how this should be performed. With most of Africa being drylands [1], the majority of restoration activities and interventions are designed to help increase climate resilience of both agro-sylvo-pastoral systems and sequester carbon, but also enable the creation of income, green jobs and tackle food insecurity and malnutrition. Africa's Great Green Wall (GGW), an initiative launched by the African Union in 2007 in response to the worsening of land degradation around the Sahara [2], targets the restoration of 100 million hectares by 2030. All of the defined national action plans and implementation interventions prioritised land restoration as the major operation of the GGW programme.

The core area of the GGW in need for restoration was estimated to be 166 million hectares in the arid and semi-arid zones around the Sahara, which equates to 10 million hectares to be restored per annum in order to reach the Sustainable Development Goals by 2030 [3]. This is an ambitious goal, which necessitates reliable approaches and methodologies for the implementation, reporting and accountability of each single intervention.

However, evaluating the successes, failures or impacts of such restoration interventions on the ground and over time remains challenging due to the lack of standardised and affordable methodologies. Poor quality monitoring practices, particularly in the long-term and over large scales, contributed to scarce understanding of the efficacy of restoration efforts [4]. Tools have been lacking to the point that it is impossible today for responsible institutions, nationally and internationally, to accurately report on what has been achieved on land restoration so far. Such monitoring tools can be used to support reporting on the progress on restoration interventions. For instance, a recent report concluded underwhelming estimates with a very large gap between 4 and 18 million hectares implemented in GGW Sahel so far [5].

Remote sensing provides spatially consistent, cost-efficient monitoring capable of accurately tracking progress on the GGW and other restoration interventions globally. Freely available satellite imagery from the public archives of USGS' Landsat [6,7] and the European Commissions' Copernicus programme (EC) allow for spatio-temporal consistent monitoring of the Earth's surface up to 10 m of spatial resolution. Up to now, remote sensing-based approaches of monitoring restoration interventions focus on the medium to long-term impact by using vegetation indices derived from optical remote sensing in combination with statistical assessments [8,9]. The use of more advanced biophysical parameters such as solar-induced chlorophyll fluorescence and vegetation water content from active microwave remote sensing data has been proposed as well [10]. However, all of those methods have in common that they are only effective after vegetation growth sets in and green plant matter is present. The time-lag between the actual intervention and the greening of planted vegetation often exceeds the project's lifetime. Therefore, there is a need for methods that are capable of immediately verifying intervention actions in order to spur investments into land restoration [11]. The use of very-high resolution satellite imagery (<5 m spatial resolution) is a valid option, as scars are clearly visible within the scenes. However, the data come at a cost, and scene coverage is limited to small areas so that multiple scenes need to be acquired for large-scale verification. FAO's Collect Earth software allows for assessing very high-resolution data for free through Google Maps and Microsoft Bing Maps, but it does not provide any control over the acquisition dates, so that interventions might be missed.

In this work we present an innovative and cost-efficient approach for near-real-time monitoring of large-scale restoration interventions in the Sahel using high-resolution Synthetic Aperture Radar (SAR) imagery from the Copernicus Sentinel-1 mission [12]. The method capitalises on the well-known sensitivity of SAR data to soil roughness [13,14], which was successfully demonstrated for Sentinel-1's C-band SAR over semi-arid areas [15]. The SAR satellites carry an antenna to actively monitor the Earth's surface by transmitting and receiving electromagnetic energy in the microwave portion of the electro-magnetic spectrum. The physical properties of the wave interaction with the Earth's surface are fundamentally different compared with the visible and infrared spectrum used by optical sensors.

An advantage of this technology is that it can acquire images independent of day- and night-time as well as cloud coverage. The SAR image data represent the energy scattered back by the Earth's surface. The intensity of the backscatter depends on both the sensor and surface parameters, which are characterised by the di-electric constant as well as the micro- and macro-geometry of the observed object. In practical terms, the di-electric constant of natural objects is mainly influenced by its moisture content. The macro-geometry relates to the terrain and the orientation of an entire pixel towards the instrument. Instead, the micro-geometry depends on the size, shape and orientation of the scattering elements within a radar pixel and determines the pixel's dominant scattering mechanism [16].

Flat surfaces such as paved roads, calm water or smooth soil surfaces normally appear as dark areas within the images as most of the incident radar pulses are reflected away from the sensor in a specular manner. The backscatter of rougher surfaces is scattered randomly until it reaches a diffuse Lambertian distribution, leading to brighter pixels. Man-fabricated structures such as buildings and bridges usually have the highest backscatter values as

their structure and orientation causes the so-called double-bounce scattering effect, where most of the transmitted energy is scattered back to the sensor.

## 2. Materials and Methods

### 2.1. Study Sites

The sites for land restoration interventions in Burkina Faso, Niger, Nigeria and Senegal were selected in the Great Green Wall areas, where mechanised land ploughing started with support from the FAO's Action Against Desertification in 2016. These sites in the Sahel are characterized by a gradient of precipitation from the desert in the north with less than 100 mm rainfall per year to the Sudanian zone in the south with 600 mm/year. The growing season in these areas is monomodal with rains falling between mid-June and October, with maximum rainfall occurring in August. In the Sahel, this type of land preparation mimics the traditional ''half-moons'' or micro-dam trenches dug in the dry season to collect rainwater during the rainy season for planting, benefitting subsequent growth of biomass (Figure 1). In consultation with village communities, different localities with degraded lands were identified to initiate restoration, ranging mostly from bare lands to sylvo-pastoral degraded lands [17]. A total of 111 plots in the four countries were investigated as study sites (Figure 1).

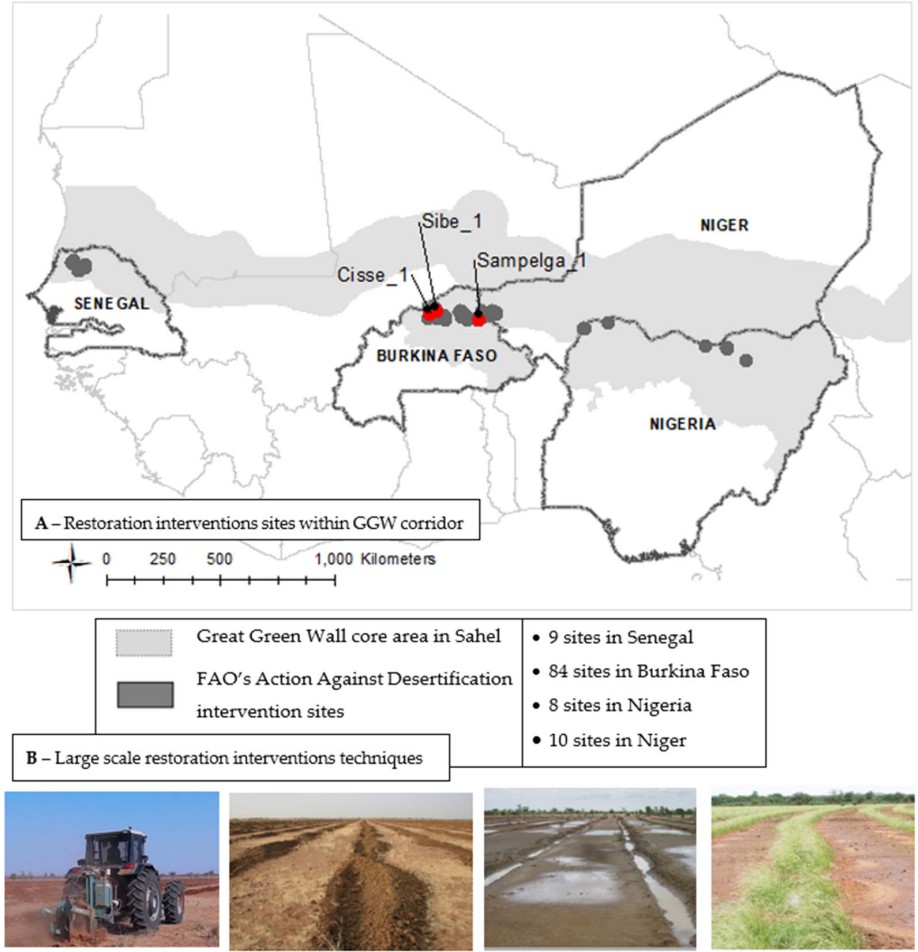

**Figure 1.** (**A**) Map of the 111 large-scale restoration intervention sites in the Great Green Wall in Burkina Faso (with three sites highlighted and further described in sections below), Niger, Nigeria and Senegal, which were assessed using radar detections. (**B**) The techniques used for large-scale restoration interventions on the ground from mechanised land preparation for soil permeability and rainwater harvesting and beginning of seedlings/biomass growth in the field.

Detailed geographic polygons, with a boundary denoted with GPS coordinates of each plot of restoration interventions, were collected through field visits by operators. These geographic data were used to compute and delineate each plot of intervention areas.

## 2.2. Verification of Mechanised Ploughing Using SAR Data

The C-band SAR instrument has ideal properties to detect soil disturbances due to mechanised ploughing interventions over bare soils in drylands. The sensitivity to surface roughness is a function of the sensor's wavelength and the incidence angle. The appearance of smooth and rough surfaces can be determined by the modified Rayleigh criterion [14], where a surface is considered smooth, when:

$$h_{RMS} < \frac{\lambda}{25 \, cos \, (\theta \text{inc})} \tag{1}$$

and a surface is considered rough, when:

$$h_{RMS} > \frac{\lambda}{4 \, cos \, (\theta \text{inc})} \tag{2}$$

where $h_{RMS}$ is the root mean square height of the surface, $\lambda$ (lambda) is the instrument's wavelength, and $\theta$ (theta) is the incidence angle. In the standard interferometric wide swath mode of Sentinel-1 the nominal incidence angle ranges from 30 to 45° and the instrument C-band antenna operates at a wavelength of 54 mm. A perfect specular scattering takes place at 1.5–1.8 mm of $h_{RMS}$, whereas 9.5–11.5 mm of $h_{RMS}$ reflect a rough surface. While the roughness of untreated bare soil usually is below this value, mechanised ploughing affects a considerable part of the radar pixel where this value is exceeded, thus leading to a considerable increase in backscatter.

This assumption is the basis for our method. A key issue that needs to be considered is the backscatter sensitivity to soil moisture. Within the Sahel, the backscatter signature of bare soils typically exhibits a seasonal pattern of backscatter due to changes in soil moisture related to alternating wet and dry seasons, thus ranging from low to medium-high values. However, as ploughing interventions take place right before the wet season, an irregular increase in backscatter can be observed in the time series, as well as in the spatial domain with respect to neighbouring areas.

To identify the ploughing interventions a dedicated script was developed to interactively browse time series data and multi-temporal composites of Sentinel-1 data using Google's Earth Engine platform [18]. On the platform, the Sentinel-1 intensity data are already pre-processed by thermal noise removal, radiometric calibration to the normalized backscatter coefficient sigma nought, terrain correction and the conversion from power to the decibel scale. The imagery has a nominal resolution of 10 m and the revisit time amounts to 12 days per orbit for the Sahel zone. This allows to determine the ploughing interventions in near-real-time at a spatial detail that allows for an accurate estimate of land area affected.

## 2.3. Medium to Long-Term Monitoring of Vegetation Growth

The type of vegetation planted are typical Sahelian multi-purpose species, which are useful and beneficial to rural communities, and are prioritized by them for their livelihoods and because they are resilient to such dry agro-sylvo-pastoral systems and landscapes. The application of these technologies and prepared lands must complement fieldwork with communities on what species to plant. Across all the four countries, over 150 species were identified as useful by communities for food, feed, human and veterinary health or cultural usages following consultations, and 110 of these species were prioritized and planted to initiate degraded land restoration.

There are different approaches to map out the extent and conditions of lands and remote sensing provides meaningful proxies, which are used to assess land conditions and monitor changes in land conditions. The normalized difference vegetation index (NDVI)

is commonly used and often these satellite-based observations are often combined with in-situ measurements. To comparatively assess vegetation growth, the data are processed from the distinct colours (wavelengths) of visible red and near infrared portions of the electromagnetic spectrum. As an indicator but not a direct measure of vegetation biomass, NDVI was used to determine the density of green plants on each plot of land under restoration and the monthly and yearly data were compared with monitor biomass growth in all the planted plots.

The trajectory measures the rate of change in primary productivity over time. A linear regression at the pixel level was computed to identify areas experiencing changes in primary productivity for the period under analysis. A Mann–Kendall non-parametric significance test is then applied, considering only significant changes that show a $p$-value $\leq 0.05$. Positive significant trends in NDVI indicate potential improvement in land conditions, and negative significant trends indicate potential degradation. To consider the effects of climate on the NDVI time series, the residual trend (RESTREND) approach was adopted. RESTREND is defined as the fraction of the difference between the observed NDVI and the NDVI predicted from climate data [19].

## 3. Results

Restoration interventions covered 84 sites in Burkina Faso, where the mechanised ploughs were first deployed, 10 sites in Niger, 8 sites in Nigeria and 9 sites in Senegal, all in the Great Green Wall areas of those Sahel countries (Figure 1). There are three specificities for the Sahel environment and context of the four countries including (i) seasonality, with a dramatic decrease in the radar signal, which was observed in SAR temporal profiles at the end of the dry season, i.e., in April and May, indicating a sudden drop in soil moisture. (ii) The contribution of trees and shrubs remained low but constant throughout the year, while annual vegetation and bare lands were the two main parameters influencing the temporal evolution of the radar signal. (iii) Bare land contribution dominated during the whole annual cycle, except when a high fallow production was observed, due to the low annual vegetation cover fractions over the studied areas.

### 3.1. Analyses and Interpretation of Radar Images

The Earth Engine script was used to select and filter SAR images for the areas of interest and extract average decibel values (conversion from backscatter coefficient to dB) in the restoration plot boundaries. As presented in Figure 2, the alternating seasonal pattern of the backscattering signal, which was stable during dry seasons, spiked in rainy seasons when the soil is moist and vegetation develops, reaching a maximum at the peak of biomass production. The peak then decreased when the vegetation senesced, and the soil dried out in dry seasons. Land ploughing operations that occurred on the ground were observed and recorded as unexpected soil disturbance in such pattern. In dry seasons (2015−early to 2020), the monthly average SAR backscatter for the Sampelga site showed a normal pattern of values between $-18.23$ and $-19.61$ dB, with exceptional values between $-13.75$ dB (March 2017) and $-16.82$ dB (June 2017) due to land disturbances. Similar patterns were observed for the Sibe site, with values of normal dry seasons of decibels between $-19.54$ and $-21.30$ and the exception period in February to June 2017 following land preparation activities, in which the average backscatter values were between $-15.61$ (March 2017) and $-18.76$ dB (June 2017). However, no such changes were observed in the normal average SAR backscatter values for the Cisse site, which remained between $-16.36$ and $-17.95$ dB with no particular evidence of unexpected alterations.

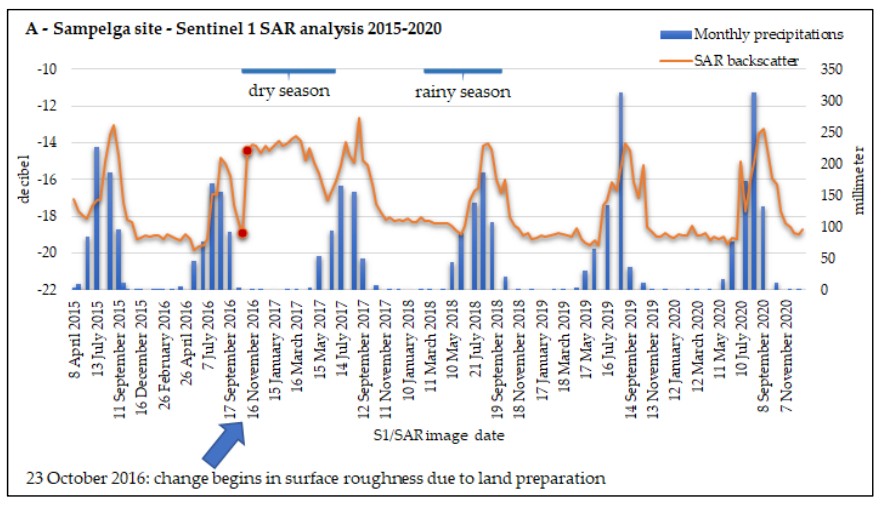

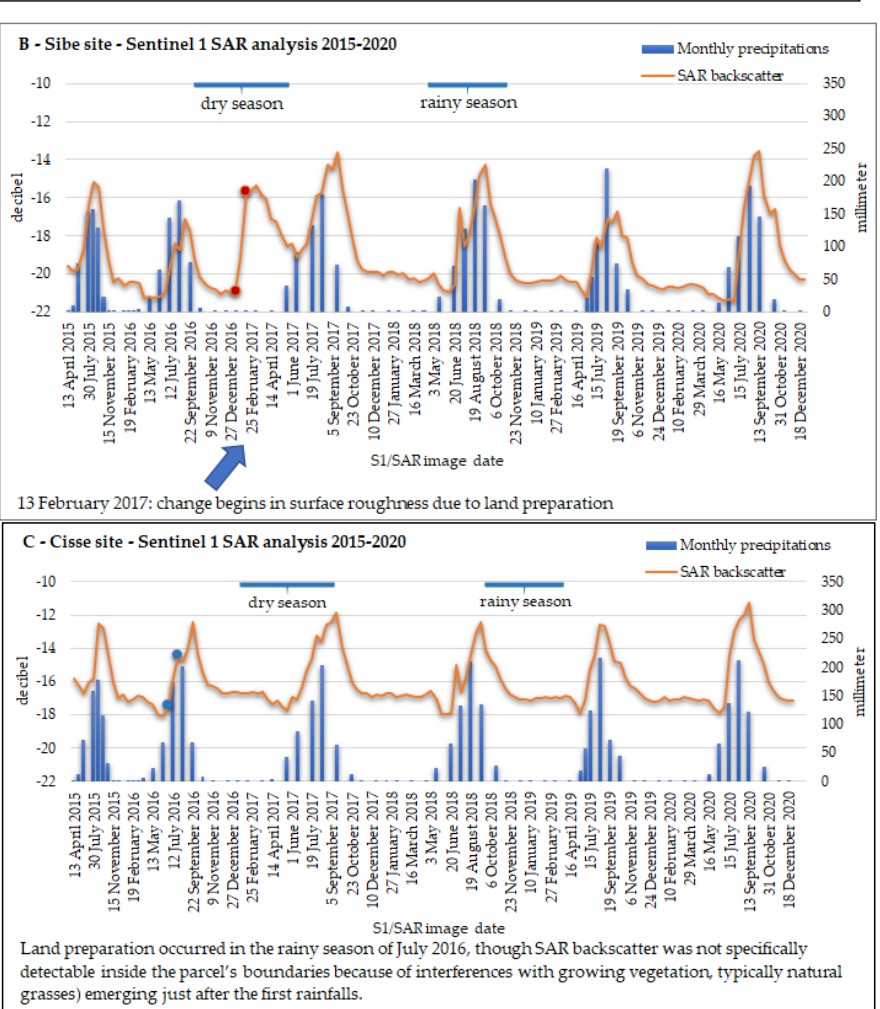

**Figure 2.** Radar detection with the time series profile of mean decibel value of restoration plots ploughed in dry and wet seasons in northern Burkina Faso, combined with total monthly precipitations. (**A**) At the onset of the dry season (October 2016) dates and pattern on the ground were clearly detected in Sampelga village (87.2 ha). Panels in GEE application (see GEE code link at https://code.earthengine.google.com/0dd6c148e4a8a56b8c3a5f59849faea7, accessed on 6 September 2021) for selecting and displaying images of the GPS delineated plot (in red line). (**B**) Similar detection was obtained for the site of Sibe (250.4 ha) mid-dry season (January 2017). (**C**) No detection was made for the site of Cisse (152 ha), because it was ploughed in the wet season (July 2016) when surface roughness also increases by new vegetation.

The same script plotting the time series analysis generated a graphic of multi-temporal RGB image composite for two different dates of SAR image acquisition. The first image is assigned to the red channel, while the blue and green channels are used for the second image. The images of before/after ploughing were selected by interacting with the application identifying the before/after dates and using sliders to navigate the time series. Based on the profile of average radar response in the plots (Figure 2) specific dates were plotted through the same script, so that images were observed in before and after initial interventions. A typical signature of an intervention shows the restoration area in turquoise (Figure 3A,B), while the neighbouring areas of bare soil remain dark. The field-collected boundaries of the restoration sites were uploaded and used to assist the interpretation. These analyses provided evidence of periods of occurrence and extents of land ploughed, as a validation of the onset of restoration activity. Multi−temporal RGB composites of SAR data were used to highlight abrupt increases of the backscattering at pixel level between the following dates, here older (before) in red and newer (after) in green/blue. Land preparation for restoration activities in dry seasons (Figure 3A,B) generated an increase in surface roughness which was captured by SAR images and represented in turquoise by this RGB combination. Land preparation in wet seasons made detection harder because of interferences of new vegetation (grasses) which equally produced surface roughness (Figure 3C).

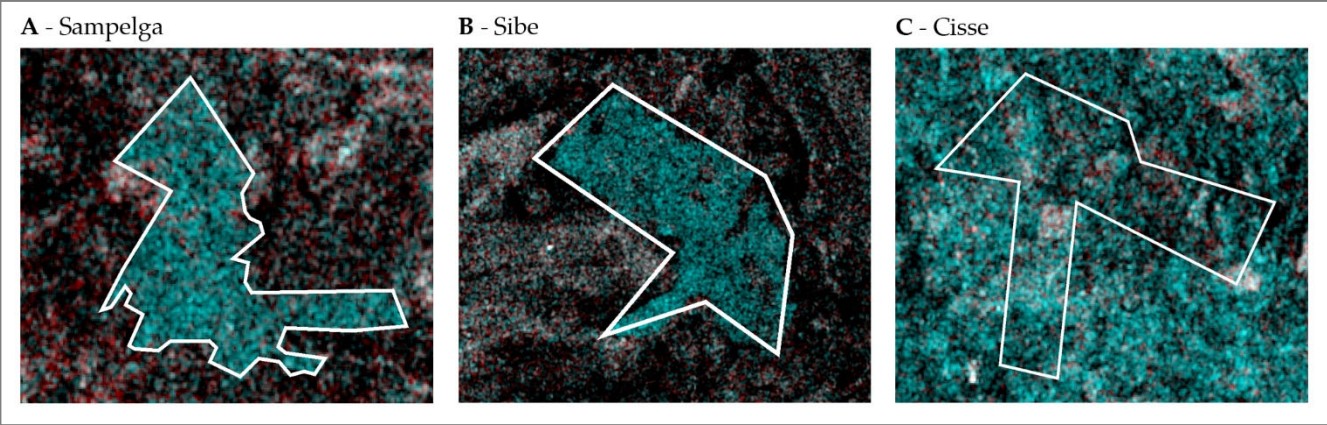

**Figure 3.** Multi-temporal RGB composites of Sentinel-1 SAR, VV backscatter. (**A**) Sampelga site: composite of images of 23 October 2016 (Red) and 16 November 2016 (Green/Blue). (**B**) Sibe site: composite of images of 8 January 2017 (Red) and 13 February 2017 (Green/Blue). (**C**) Cisse site: composite of images of 6 June 2016 (Red) and 6 August 2016 (Green/Blue), coinciding with the rainy season.

### 3.2. Analyses of the Great Green Wall Restoration Plots

A total of 7111 ha of village lands in 111 plots were ploughed in 2016–2017 in Burkina Faso, Niger, Nigeria and Senegal and were comparatively analysed through radar detection and GPS delineated boundaries (Table 1). The majority, 60% of the area (4248.1 ha) could be detected by the radar imagery after ploughing, while 40% remain undetected. The radar-detected areas were either similar, larger or smaller than the field areas collected by GPS and were thereafter appropriately adjusted for the records and reporting (Table 1).

**Table 1.** Land restoration plots assessed through radar detection in the Great Green Wall areas of four Sahel countries.

| Peak Radar | Burkina Faso | Niger | Nigeria | Senegal | | Total | |
|---|---|---|---|---|---|---|---|
| Detected plots (count) | 49 | 8 | 0 | 1 | 58 | 52% | 111 Plots |
| Undetected plots (count) | 35 | 2 | 8 | 8 | 53 | 48% | |
| Areas detected (ha) | 3093.6 | 907.1 | 0.0 | 247.4 | 4248.1 | 60% | 7111.3 ha |
| Areas undetected (ha) | 782.0 | 165.9 | 436.2 | 1479.2 | 2863.3 | 40% | |

It was observed that all 53 plots which could not be detected by radar imageries were ploughed in July or October during the rainy season. The remaining 58 detected plots were prepared in the dry season mostly between December and March. For some of the detected plots the accurate dates could not be specified for land preparation activities whatsoever according to the typical signature of an abrupt change in surface roughness (see Figure 2C Cisse). Field visits revealed that most of these plots were already covered by substantial vegetation or were located on steep slopes.

### 3.3. Monitoring of Biomass Growth in the Restoration Plots

All the lands ploughed were planted with a mix of woody and fodder herbaceous species through mainly direct sowing of seeds complemented with some nursery seedlings, in June at the onset of the rainy season (see Table 2 [17]. Trained village technicians were looking after the plots and collecting data in the field and digital assessments were performed with changes in NDVI–Landsat (30 m).

**Table 2.** Major species prioritized and planted for land restoration by communities in the Great Green Wall. About one-third of the species were grasses used as fodder, food and/or feed for people and the livestock and were directly sown in mix with woody species.

| Selected Priority Native Species | Life Form | Main Uses by Communities | Average Seed Germination (%) | Planted Form in the Restoration Plots |
|---|---|---|---|---|
| *Acacia nilotica* | Shrub | Gum, fodder | 100 | Seeds and seedlings |
| *Acacia senegal* | Shrub | Gum arabic, bees, forage | 100 | seeds and seedlings |
| *Acacia seyal* | Tree | Gum, fodder | 95 | Seeds and seedlings |
| *Acacia tortilis* | Shrub | Gum, fodder | 100 | Seeds and seedlings |
| *Adansonia digitata* | Tree | Food, medicine | 80 | Seeds and seedlings |
| *Alysicarpus ovalifolius* | Grass | Feed, fodder | 60 | Seeds (10 kg/ha) |
| *Andropogon gayanus* | Grass | Roofing, forage | 100 | Seeds (5 kg/ha) |
| *Anogeissus leiocarpa* | Tree | Wood, medicine, dyeing | 90 | seedlings |
| *Balanites aegyptiaca* | Tree | Food, oils, medicine, fodder | 100 | Seeds and seedlings |
| *Bauhinia rufescens* | Shrub | Fodder, fence, rope | 100 | Seeds and seedlings |
| *Bombax costatum* | Tree | Food, fodder, mattress | 58 | Seeds and seedlings |
| *Ceiba pentandra* | Tree | Wood, food, mattress | 98 | seedlings |
| *Cenchrus biflorus* | Grass | Fodder | 35 | Seeds (5 kg/ha) |
| *Combretum glutinosum* | Shrub | Fodder. Wood, medicine | 95 | seedlings |
| *Combretum micranthum* | Shrub | Fodder, food, medicine | 100 | seedlings |
| *Cymbopogon giganteus* | Grass | Medicine, beverage, pesticide | 56 | Seeds (5 kg/ha) |
| *Detarium microcarpum* | Tree | Food, fodder | 70 | Seeds and seedlings |
| *Digitaria exilis* | Grass | Food, feed | − | Seeds (0.5 kg/ha) |
| *Digitaria horizontalis* | Grass | Food, feed | − | Seeds (0.5 kg/ha) |
| *Eragrostis tremula* | Grass | Fodder, forage | 75 | Seeds (0.5 kg/ha) |
| *Euphorbia balsamifera* | Shrub | Living Fence, medicine | 25 | Seedlings |
| *Faidherbia albida* | Tree | Fodder, medicine, wood | 100 | Seeds and seedlings |
| *Grewia bicolour* | Shrub | Food, medicine, feed | 3 | seedlings |
| *Khaya senegalensis* | Tree | Wood, medicine, pesticide, fodder | 100 | seedlings |
| *Lannea microcarpa* | Tree | Food, rope | 80 | seedlings |
| *Panicum laetum* | Grass | Food, feed | 20 | Seeds (5 kg/ha) |
| *Parkia biglobosa* | Tree | Food, medicine, bees | 100 | seedlings |

**Table 2.** *Cont.*

| Selected Priority Native Species | Life Form | Main Uses by Communities | Average Seed Germination (%) | Planted Form in the Restoration Plots |
|---|---|---|---|---|
| *Pennisetum pedicellatum* | Grass | Fodder | 100 | Seeds (0.5 kg/ha) |
| *Prosopis africana* | Tree | Food, medicine, wood | 100 | seedlings |
| *Ptérocarpus erinaceus* | Tree | Wood, medicine, bees | 95 | seedlings |
| *Sclerocarya birrea* | Tree | Food, feed, wood | 80 | seeds and seedlings |
| *Senna tora* | Grass | Fodder | 30 | Seeds (5 kg/ha) |
| *Strychnos spinosa* | Shrub | Medicine, pesticide, fodder, wood | 89 | seedlings |
| *Stylosanthes hamata* | Grass | Fodder | 90 | Seeds |
| *Ziziphus mauritiana* | Shrub | Food, fence, medicine | 87 | Seeds and seedlings |
| *Zornia glochidiata* | Grass | Fodder | 55 | Seeds (5 kg/ha) |

Both seeds and seedlings from prioritized multi-purpose trees, shrubs or grasses, were planted in the ploughed plots to initiate their restoration and rehabilitation of degraded agro-sylvo-pastoral systems. A mixture of a minimum of 10 species, often 6 woody and 4 grass, were planted per hectare, maximizing social interests and ecological functions and resilience on the ground. Although nursery seedlings tended to have a greater survival rate in the first couple of years, there was no significant difference between 3rd year seedlings either from direct sowing of seeds or from nursery seedlings. Direct sowing was more cost-effective, especially in the case of large-scale restoration where significant surface areas were targeted and large quantities were planted (Table 2).

Results showed NDVI variations comparing the monthly averages over a 10 year period before interventions (2010–2014) and after interventions (2015–2020). The trends were polynomial and presented clear vegetation gains for the monthly aggregates over the last 2 years (2018–2020), which corresponds to the time of the third or fourth rainy season after planting (Figure 4). For instance, vegetation gains with NDVI values of 0.15 in 2017 increased up to 0.2 in 2020 toward the end of the data series with higher peaks in months of the latest years. This was used as a tool to assess plots with or without increase in vegetation, demonstrating the long-term success or failure of restoration interventions.

Planted plots were regularly monitored, combining field data collected and digital monitoring for accurate reporting on these restoration interventions. With NDVI–Sentinel-2/Copernicus (10 m), visual representations of restoration plots were used for qualitative (greening patches) and quantitative (biomass increases) assessments of the planting success or failure. Examples for three sites in Burkina Faso are presented in Figure 3 where two seasons of collected data were compared at the end of dry season, i.e., in May 2020 and 2021. The focus was on a dry season period when there was hardly any grass in the field, to highlight mainly the survival of mainly established woody seedlings. From the observations in 2020, Sampelga and Sibe plots seemed more degraded, i.e., less green patches within the boundaries than the Cisse plot with existing green patches (woody vegetation). However, both the Sibe and Cisse plots showed increases and intensifications of green patches of vegetation in 2021 compared with the situation in 2020. This was probably due to the successful growth of 3 years old woody seedlings. Whilst within the Sampelga plot hardly any greening was observed in 2021, due probably to the fact that those seedlings had not yet fully established for the same period, as also observed in the field data (not shown). Visual and digital monitoring were systematically applied as a standardised method to follow-up all restoration interventions and contrasted with field qualitative and quantitative data (i.e., proportions of germinated seeds per species, establishment/survival of seedlings, soil vegetation coverage and annual growths of planted species).

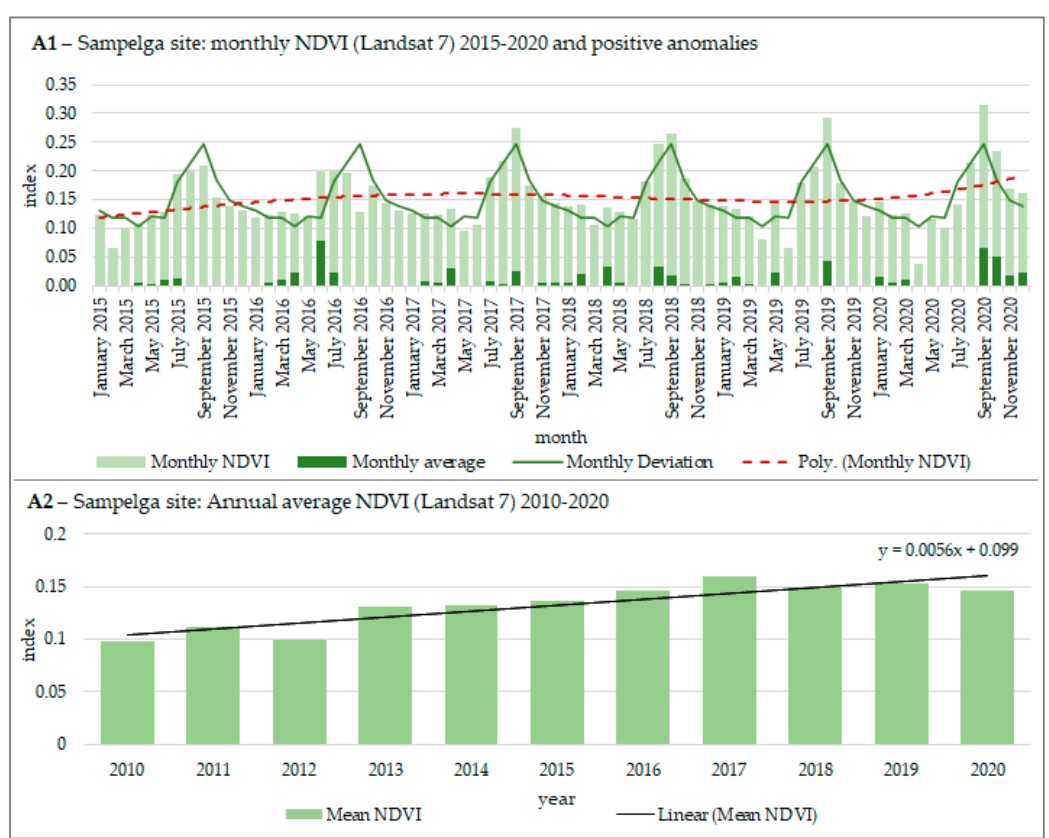

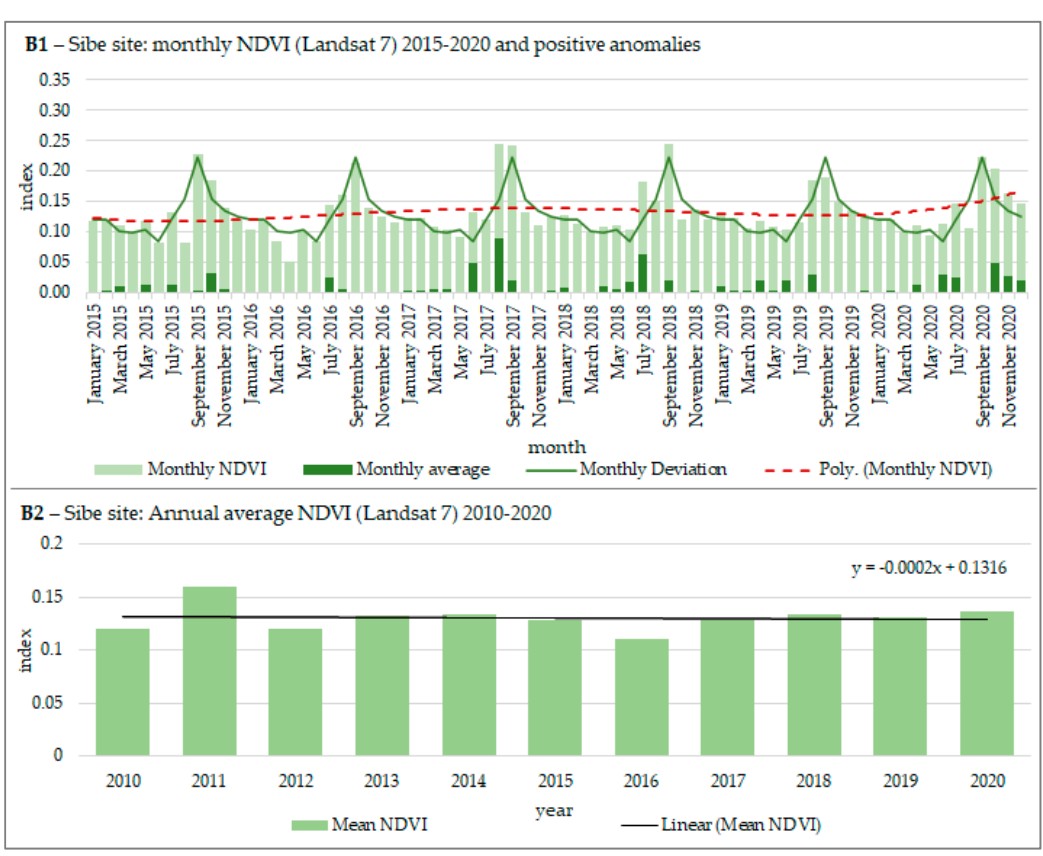

**Figure 4.** *Cont.*

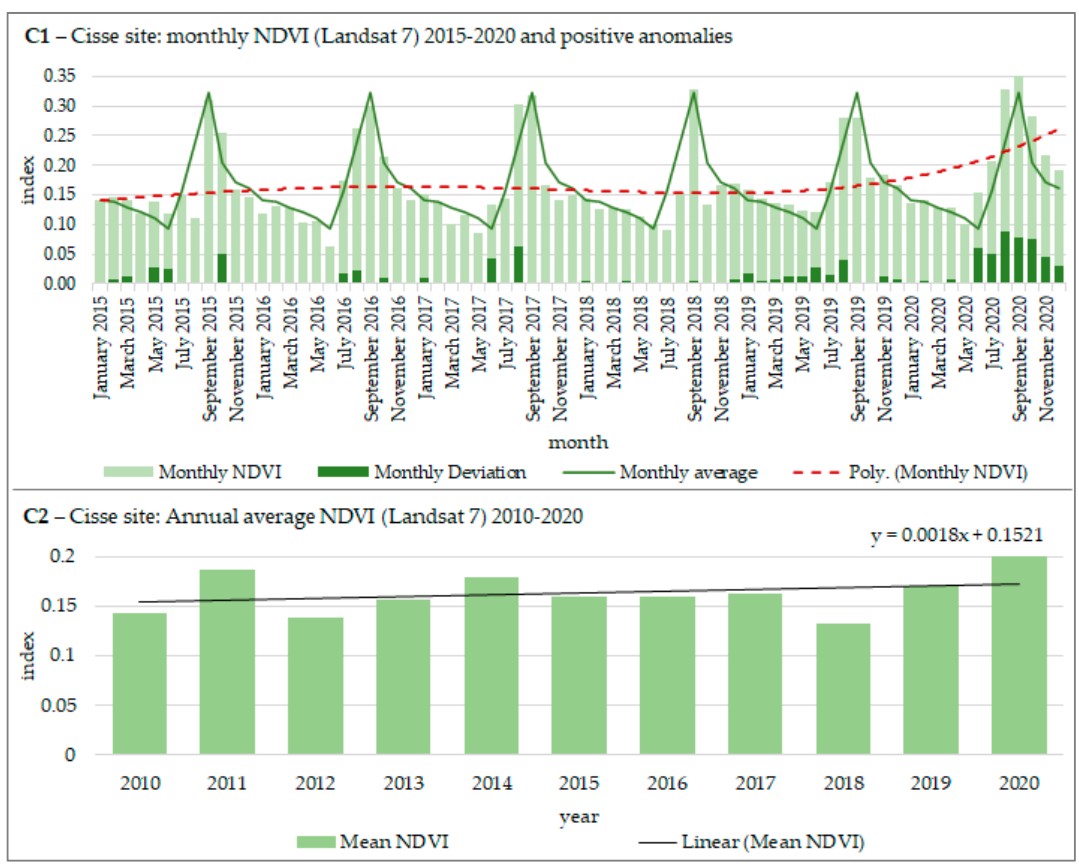

**Figure 4.** Digital monitoring of restoration plots in Sampelga (**A1**), Sibe (**B1**), and Cisse (**C1**), Burkina Faso, with NDVI−Landsat (30 m) variations comparing the monthly and annual averages for the period before interventions (2010–2014) and after interventions (2015–2020). The linear and polynomial trendlines show clear gains toward the end of the data series with higher peaks in months of latest years (**A2,B2,C2**), demonstrating increase in vegetation, with values of 0.15 in 2017 up to 0.2 in 2020 for the Cisse site (**C2**). The recurrent detection of negative deviations from average in rainy months is due to lack of data (cloudy images out of 2 images per month).

## 4. Discussion

Globally, large-scale land and landscape restoration recently gained national and international attention to adapt to the changing climate and mitigate its effects with renewed commitments to plant billions of trees and restore up to 1 billion hectares in the decade 2021–2030. These restoration interventions are carried out to increase biomass production, improve vegetation cover and land productivity for livelihoods, biodiversity and carbon sequestration. Therefore, a measurement, reporting and verification system must be in place to help assess and track progress. Our results demonstrated that large-scale land restoration interventions can be independently and comparatively assessed within the very first periods of ground operations through to the build-up of biomass in standardised and affordable methods.

In this study, large-scale land restoration, implemented under the Great Green Wall initiative, was assessed using Sentinel-1 SAR intensity data and capitalized on its sensitiveness to soil disturbances to detect interventions by mechanised ploughing of degraded lands. These investigations required interpretation and application of free available online tools that were combined and matched with field data to confirm or correct the delineations of sites under restoration interventions. This approach was a very helpful and handy management tool and supported accurate regular reporting on progress. In the context of the Great Green Wall programme, as in any other land restoration intervention, there are cost implications in every single hectare planned to be restored. Quantitative/qualitative field data, success or failure should be evidenced from the field and in visual representations. In

addition, if not checked and reported independently, projects might be paying too much or too little for what was not accurately effective on the ground.

Ploughed plots were planted with a mixture of grass and woody species that were selected not only to initiate land restoration and the rehabilitation of the agro-sylvo-pastoral systems, but also in consideration of communities' interests and improvement of their livelihoods. Multi-purpose species with different uses and high market value were usually preferred, planted and looked after, which included highly nutritious food-source woody species such as Egyptian balsam (*Balanites aegyptiaca),* Indian jujube (*Ziziphus mauritiana*), and African baobab (*Adansonia digitata*). In combination, grass fodder species used for livestock grazing, e.g., *Alysicarpus ovalifolius, Pennisetum pedicellatum, Senna tora* and *Zornia glochidiata*, were also sown, which was established quickly in the plots within the 1st year of planting. The economic benefits and the incentive for maintenance of plots and plants include for example, revenues generation from fodder species during the first years of planting by communities [20,21].

The follow-ups of restoration interventions were performed through measurements of the normalized difference vegetation index (NDVI) after land was prepared for soil permeability and rainwater harvest, and thereafter planted with the right resilient native species in the right places [21]. Biomass monitoring provided a clear picture of greening (or not) in the plots under restoration. The technique of images combined to provide a multi-temporal colour composite image of the areas was especially useful in detecting land cover changes over the period of image acquisition. Areas were distinctive where change in land cover occurred as colourful (green) patches in the image. In a comparative assessment of these ploughed lands, a good match was found between field data versus computerised data using Radar techniques. Analyses of the consecutive years with high-resolution imagery and the NDVI of biomass subsequent growth in the plots, and qualitative data of species planted, also show an increase in biodiversity as a minimum of 10 well-adapted native species 6 woody mixed with 4 herbaceous, were planted per hectare. It was observed that for some of these digital evaluations a 3 year period was too short to pick up the vegetation greening with NDVI−Sentinel-2/Copernicus (10 m), as shown for the most degraded lands such as in Sampelga (Figure 5). This by no means marks a failure of establishment of newly planted vegetation. Rather, it is a sign of slow growth of seedlings in those specific locations, as evidenced by the collected field data (not shown). For a robust evaluation of restoration interventions, the team used visual and digital monitoring, systematically applied as a standardised method to follow-up all restoration interventions, and complemented with field qualitative and quantitative data, including the proportions of germinated seeds per species, establishment/survival of seedlings, soil vegetation coverage and annual growths of the planted species.

However, the application of the digital methodology has limitations, with assessments showing that they did not work when land preparation occurred in the rainy season or within relatively vegetated lands, where radar detections were compromised. In those cases, detailed ground-proofing was applied for confirmations on what effectively worked. Successful radar detections were confirmed in 60% of all plots and corresponded to the majority of the intervention boundaries recorded from GPS data (e.g., Sampelga and Sibe plots). While 40% of the total plots did not show a signal, field observations and later NDVI data confirmed interventions (e.g., Cisse plot). The main reason for those false negatives were that interventions took place in the wet season, where high soil moisture leads to an increase in radar backscatter or existing vegetation was already present. The use of interferometric coherence, as proposed by Shang et al. [22], can overcome this limitation. Coherence is rather independent of intensity changes due to soil moisture and an accuracy of up to 85% for seeding-date identification was found in the change in surface structure induced by seeding or harvest operations. These data are however not accessible on Google Earth Engine, and therefore necessitates fundamental understanding of pre-processing of SAR data in addition to elevated processing and data storage costs for large-scale areas. Although the methodology needs some improvement to increase the direct detection

proportions of prepared lands, the current results obtained were a good indication of reliable digital independent assessments, which must be complemented with concrete field data. Nonetheless, the subsequent years of vegetation presence and biomass increases in planted plots through NDVI regular assessments provide an independent and standardised platform for evaluating success or failure of restoration interventions. Improvements can also be made to monitor biomass growth at a species-level, as the ability of particular species to coexist at different spatial scales was cited as critical to restoration success [4]; although, we are not there yet, and field work can complement where technology is yet to reach. Nonetheless, the scale of the restoration ambitions coupled with the urgency to reverse land degradation makes this methodology a cost-effective and replicable solution to monitor restoration outcomes.

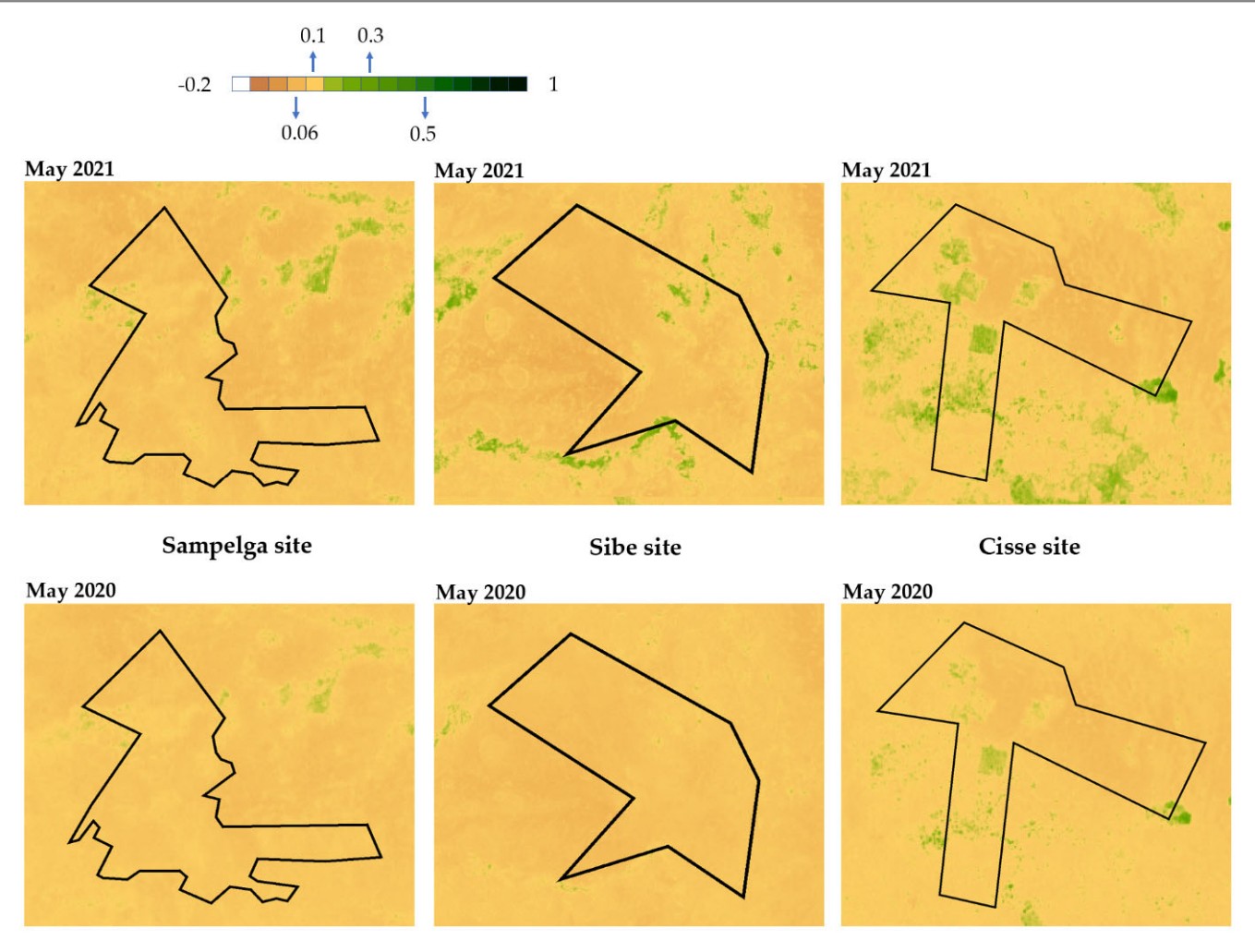

**Figure 5.** Visual representation of restoration plots in Sampelga, Sibe and Cisse, Burkina Faso, with NDVI−Sentinel-2/Copernicus (10 m). The green patches are the vegetation in the plots comparatively showing the increase from May 2020 and 2021, during the dry season.

The business case for investing in large-scale restoration is gaining momentum as a strategy for stimulating green recovery due to its potential to contribute to economic growth and employment, and provide a range of environmental and socio-economic benefits. We believe that standardised and independent monitoring methods can increase trust from investors, managers, partners, and stakeholders. Furthermore, experience from more than a decade of FAO support to Africa's Great Green Wall in the Sahel has shown that restoration interventions need to do more than plant trees if they are to stand any chance

of success. The interventions must go beyond biomass and be rooted in improving ecology, lives, and livelihoods simultaneously. Monitoring and evaluating restoration interventions through established methodologies to assess biophysical as well as socio-economic impacts have by extension accompanied FAO's interventions throughout the last decade. Alongside the progress, FAO has made to fine-tune and harness technology to monitor biomass change, tailored socio-economic assessments have equally been a core part of its monitoring and evaluation assessments. The Sustainable Livelihoods Framework developed by the United Kingdom's Department for International Development [23], FAO's food insecurity experience scale (FIES) [24] and targeted questions regarding livelihoods, food consumption and the use of native species were used to monitor and tailor interventions in the field. Future monitoring and evaluation need to combine both biophysical as well as socio-economic evaluation, which can further support decision-making on land use as well as associated challenges which need to be tackled simultaneously such as food insecurity and poverty.

Land restoration moreover is a long-term process because it needs to be embedded in the local agricultural practices and culture and tailored to the ecological features of different sites [17,20,25–27]. Restoration pledges have proliferated in recent years, and a plethora of initiatives and platforms have developed to support the ambition, funding, implementation, and monitoring of those efforts. As a result, harmonizing efforts to monitor, track and report progress on land and landscape restoration will improve coordination and consistency, resulting from close collaboration with implementers, stakeholders, donors and partners and will increase transparency and the impact of restoration initiatives. Innovative monitoring approaches combining field data and remote sensing data to provide a standardised and objective assessment of restoration interventions should be used and proposed for large-scale restoration initiatives. Comprehensive and user-friendly tools for monitoring land restoration interventions from land preparation to biomass growth are essential and have the potential to assess land use, land use change, natural disasters, sustainable management of scarce resources and ecosystem functioning. The existing tools and methodologies we developed enable non-remote sensing experts to digitally assess a large number of sites per day. FAO developed Open Foris Collect Earth tools in collaboration with Google Earth Engine and made it available to any users to use it for land monitoring and to substantively improve our collective understanding of the world's land use and land cover [28]. Through such an approach of independent and comparative assessment of land restoration, more public and private sector actors will likely feel comfortable in investing in restoration as a part of their zero-net climate mitigation and go beyond measuring biomass only, with long-term benefits for both the health of people and ecosystems.

## 5. Conclusions

Lands ploughed in the rainy season did not show clear radar detections (e.g., Cisse site) compared with lands that were prepared in the dry season due to interferences with emerging vegetation (grasses) and/or soil moisture (e.g., both Sampelga and Sibe sites). Combining assessment data, i.e., digital, visual (Figures 3–5) with field data, it was observed that 3 years of restoration results showed, for relatively less degraded sites and localities such as the Cisse site, a more successful re-greening due to planting. Their latest 2020 average vegetation indexes increased and intensive green patches were seen in visual representations. While in the same timeframe, similar observations were not yet a clear-cut for more degraded lands. There was a slight increase in the Sibe site but hardly in the more degraded Sampelga site, although its monthly average increased. The differential growth rate in the field followed similar trends with higher established vegetation in Cisse than in Sibe and Sampelga sites, respectively. These visual and digital monitoring approaches were systematically applied as a standardised method to follow-up all restoration interventions, complementing qualitative and quantitative field data (i.e., proportions of germinated seeds per species, establishment/survival of seedlings, soil vegetation coverage and annual growths of planted species).

This work demonstrates that restoration interventions in arid to semi-arid landscapes can be independently assessed by remote sensing methods throughout all phases, and that the methodology to track biomass growth from the onset of interventions is replicable elsewhere, if technology is supported with leg work in the field. Initial soil disturbances due to ploughing and time of occurrence were detected using Sentinel-1 radar imagery, and subsequently, time-series of NDVI derived from high-resolution imagery allowed to track the increase in biomass and verify the long-term impact of restoration interventions. Results for the Great Green Wall in Burkina Faso, Niger, Nigeria and Senegal showed the geographical transferability of the method within dryland regions of the Sahel. This technological monitoring approach for evaluating restoration success (or failure) in the drylands with visual representations is critical in accurate reporting, accounting and payment for large-scale restoration interventions. It can likely bring more public and private sector actors to feel comfortable in investing in restoration as a part of their zero-net climate mitigation.

**Author Contributions:** M.S. and A.M. conceived and designed the research, collected and analysed the data, A.V. contributed to the methodology and editing of the manuscript. All authors have read and agreed to the published version of the manuscript.

**Funding:** This research was funded by the European Union, through grant number GCP/INT/157/EC.

**Institutional Review Board Statement:** Not applicable.

**Informed Consent Statement:** Not applicable.

**Data Availability Statement:** The study did not report any data.

**Acknowledgments:** Thanks to Alfonso Sanchez for supporting the development of the script in Google Earth Engine and to Giulia Muir for editing the manuscript. This work was conducted under Action Against Desertification (AAD), an initiative of the Organisation of Africa, Caribbean, and Pacific States (OACPS), implemented by FAO and funded by the European Union (GCP/INT/157/EC). The authors are grateful to the many rural communities and technicians, who support and contribute to the programme, and to the AAD team, including local enumerators, who facilitated data collection in the project areas.

**Conflicts of Interest:** The authors declare no conflict of interest.

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
