# Peer review of "Monitoring Large-Scale Restoration Interventions from Land Preparation to Biomass Growth in the Sahel"

_remotesensing, doi:10.3390/rs13183767_

Round 1

Reviewer 1 Report

The paper proposes a methodology based on SAR and optical data time series to address monitoring land interventions and restoration. The study is novel and is scientific accurate. The paper has merit for publication in Remote Sensing journal following major revisions. See below my general comments:

1) Methods: describe the data better. For example, it may be missing some pre-processing, was speckle filter applied to SAR data, time period analyzed with remote sensing imagery, does not mention which optical satellite data was used to retrieve NDVI, EVI was mentioned but not used, which type of vegetation was planted. Also, SAR section has too much basic background information on how SAR works, I suggest either removing a big part of it or moving some to Introduction.

2) Results overall: be more precise and quantitative when reporting Results and move Methods that are present in results to Methods. For example, when authors describe the Figure 2 they say the signal is low or high, but this is subjective. It is important to extract summary metrics between sites to report what is the overall statistics of the SAR signal that represent the land plowing. An example of Methods present in Results is the first sentences of Results “The radar images were interpreted and analysed”. Please take a careful look in Results and be sure there is no methodological information there.

3) Discussion: would your approach work in other areas around the world? I think a few sentences or a paragraph discussing the potential of applicability of this framework for other areas could attack a broader readership and improve citation.

4) GEE code: since the study was based on GEE, it would be useful for readers interested in applying the framework to have the code. This would likely help increase citations in the future as well.

5) Minor typo and grammar check across the document.

Specific comments:

Figure 3: could you not merge the bottom graphs into one? It would help the reader to see if there is a trend between years.

Figure 4: it was not described and discussed in the text, remove or include a paragraph discussing it. Moreover, what does this figure mean? I see a lack of green patches in 2021, does that mean the restoration was not a success e.g. vegetation did not grow?

Author Response

Review 1

1) Agreed and EVI has been completely removed as it was not used in the document. Type of vegetation planted to initiate land restoration and rehabilitation restoration has been added - L 163-169 and L2283-295 including a new Table 2 listing major species; and in the discussion, specifically in L359-370. A part of SAR section is moved to the Introduction – L86-107.

2)-We thank the reviewer for this comment. We have addressed moving results to Result section and methods to Methods section. – L151-153 and L191. Extracted summary metrics were not informative, though, such extraction of metrics was challenging as locations and conditions of the fields change from one year to another. This makes it not possible to have consistent and meaningful backscatter values of the SAR images valid everywhere.

3)- We have speculated on the potential use of our approach in other similar areas (L410-421 and L482-484), though the scope of our study areas have been clearly described in the manuscript. We believe that this would be the major interest of the paper, when published so that the techniques and approach can be of use in other areas around the world.

4)GEE code has been added – see Figure 2 – L230-231

5) Minor typos and grammar have checked all along the document and corrected. Specific comments: For more clarity, Figure 2 has been split to become two; Figure 2 and Figure 3 (L236-240). Figure 4: We merged the annual NDVI and highlighted the trend of a 10-year period (2010-2020), instead of 20 years, presenting the periods before and after interventions. Figure 5: now described and explained in – L316-331. We have further discussed it in the text – L383-393 and in the Conclusion L468-479.

Reviewer 2 Report

This is a very well written article and figures are very good.

There is one misspelling on line 234> "remaining"

Author Response

Review 2

Minor typos, misspellings and grammar have been checked all along the document and corrected. We thank the reviewer for the very positive feedback and appreciation of our piece of work.

Reviewer 3 Report

This paper examined the vegetation restoration in the Sahel after the interventions using SAR and NDVI images, which is very interesting but requires some modifications, particularly for the discussion on the spatially different characteristics of the sites.

Equations 1 and 2. Please provide the roughness equations in the correct form. Special characters look broken.

Figure 1. Please add (A) to the upper figure and (B) to the lower figure. Can you mark Sampelga, Sibe, and Cisse sites on the map?

Figure 2. Please apply the same value ranges (e.g., -10 to -22) to the y-axis for VV backscattering. Also, use the same date ranges (e.g., Jan 2016 to Dec 2020) for the x-axis.

L245. Landsat 7 images usually have stripes due to the SLC-off issue. How did you deal with the stripes on the images?

Figure 3. A and C sites had the maximum NDVI in September 2020, whereas B sites had the maximum in August and September 2017. Did you explain the reason sufficiently? Also, the value of the maximum NDVI for A, B, and C is 0.32, 0.25, and 0.4, respectively. Did you discuss the difference in more detail?

Author Response

Review 3

Done; equations 1 and 2 are now presented in the correct form.

Figure 1: component A and B have been specified and the three sites, Sampelga, Sibe and Cisse are added on the map.

Peaks in monthly values were always detected at the top of the rainy season, i.e. between August and September. These are related to and coincided with particularly favorable conditions (moisture due to rains and temperatures) for vegetation growth. However, the noticeable differences seemed small, but they are within a range acceptable for these arid or semi-arid environments. The three-year period of our study was too short as impacts of restoration interventions would take longer for improved vegetation trends and detection of vegetation growths, particularly in dry months in the Sahel. As we are continuously monitoring all the plots, applying systematically and regularly this approach, future results will show such trends of biomass increases in the plots.

Reviewer 4 Report

The study tackles an important issue and is of interest to the readers of the journal. The authors explained the problem well, nevertheless, the solution offered does not seem to be novel and the validation of restoration efforts is not clear to me. The authors explained that real-time monitoring of restoration efforts will encourage investors, nevertheless, the 40% odds of not detecting the restoration effort due to seasonality effects doesn't make the solution very appealing.

I recommend that the authors explore ways to improve the sensitivity of of the freely available GEE tools to match their particular needs. If this is not possible the authors need to explore additional data sources or methods for verification of restoration efforts that took place during the wet season. Also, the software package SNAP has a lot of potential for this kind of investigation with access to multiple SAR products with varying resolutions and freely available training materials.

The trends in vegetation/biomass indication (NDVI) is not shown for nearby sites with no restoration techniques applied to them. There might be a chance that other areas might have experienced increase in vegetation without intervention as well. Also, the comparison to other sites without intervention will show if the benefits are significant or noticeable. I am not quite sure what a change of .05 in the NDVI is significant or not (as an investor).

I understand there might not be a better direct solution to the seasonality issue. Therefore, I recommend that the authors provide more recommendations on how investors can make the best of this approach (e.g., recommend that restoration efforts are done during the dry season etc.) 

Here are some specific comments:

- Line 16 don’t start sentence with numeral (happens in the manuscript body as well)

- Line 17 7,000

- Fix Equations

- Line 137 to 143 provide references if available

- Line168 potential improvement typo

- Figure 2 needs significant improvement, the dates in panel C don't seem to be correct and (R) and (GB are not clearly explained in the caption. Y axis units are needed.

Author Response

Review 4

We thank the reviewer for these comments and suggestions. We believe that the solution we propose of combining field data complemented with digital standardized monitoring from land preparation to vegetation cover increases will be very appealing, independently verifiable and with technologies available to non-expert. To our knowledge, this is currently lacking and our approach would provide more transparency and encourage investors who often rely only on reports and/or are in limbo about success or failure of their restoration efforts.  

We do agree that the 40% odds of not detecting the restoration effort due to seasonality effects still need more investigations on improved detection options for land preparation, but in a challenging way both accessible to non-expert and freely available.

Although this was not a part of our study, we do agree with the reviewer that the comparison with the trends in vegetation/biomass indication (NDVI) for nearby sites with no restoration techniques applied might be interesting to investigate. This suggestion will be considered in our future data collections. In the context of the Sahel and dryland enviroments, a change of .05 in the NDVI is significant as these sites are severely degraded, often less fertile/productive and not even useable by farmers and completely depleted from vegetation; hence their restoration and rehabilitation.

The seasonality is linked to detection in dry season and no-detection during the wet season, of land disturbance in the restoration process. But not directly linked to restoration effort per se. Therefore, we do not agree and cannot make a recommendation that restoration efforts should occur during the dry season. The general agricultural/planting calendar in the Sahel is based on the 3-4 month rainy season (no irrigation) and lands are subsequently ploughed in the dry season to capture the maximum rainwater during that production period. Restoration efforts also follow similar trends.

Specific comments:

All typos, misspellings and corrections on equations and sentences have been addressed, with Figure 2 split to separate with Figure 3 and better explained in the revised version.

Round 2

Reviewer 1 Report

The authors did a great job improving the manuscript and in my opinion it is now in good shape for publication in the Remote Sensing journal. I do not have any additional comments. Congrats.

Author Response

We thank the reviewer for the constructive suggestions, which helped to improve the manuscript.

Reviewer 3 Report

# Figure 2. High/low is not enough for the color bar. Digits representing the yield amount (e.g., ton/ha) should be included.

# Figure 5. The same as Figure 2.

# Line 326. You should present the skeleton of each GWR model to show what explanatory variables are included (like below).

Corn=GWR(Var1, Var2, Var3, …, VarN)

Soybean=GWR(Var1, Var2, Var3, …, VarN)

Wheat=GWR(Var1, Var2, Var3, …, VarN)

# Moreover, you should present the result of OLS or MLR (multiple linear regression) to show that GWR was superior to OLS/MLR. You can use the same explanatory variables (like below).

Corn=MLR(Var1, Var2, Var3, …, VarN)

Soybean= MLR(Var1, Var2, Var3, …, VarN)

Wheat= MLR(Var1, Var2, Var3, …, VarN)

Author Response

We thanks the reviewer for the request of more quantitative data and statistical analyses in Figures 2 and 5.

Figure 2: We have now described with some digits/values that are extracted from Figure 2 and explained in the text – see Lines 222 to Lines 232.

Figure 5: We did not collect and do not have any actual biomass estimates in tons/ha for this study. Therefore, we could not work out statistics related to quantitative increases of biomass in the plots. We presented visual representations and we think that such qualitative changes should be valuable, as proxies and indicative along with increases in biomass.

Reviewer 4 Report

Thanks for addressing my comments.

Author Response

We thank the reviewer for the constructive suggestions that help to improve the manuscript.